# Thiolated Hydroxypropyl-β-cyclodextrin: A Potential Multifunctional Excipient for Ocular Drug Delivery

**DOI:** 10.3390/ijms23052612

**Published:** 2022-02-26

**Authors:** Brunella Grassiri, Patrick Knoll, Angela Fabiano, Anna Maria Piras, Ylenia Zambito, Andreas Bernkop-Schnürch

**Affiliations:** 1Department of Pharmacy, University of Pisa, Via Bonanno 33, 56126 Pisa, Italy; brunella.grassiri@phd.unipi.it (B.G.); angela.fabiano@unipi.it (A.F.); anna.piras@unipi.it (A.M.P.); 2Center for Chemistry and Biomedicine, Department of Pharmaceutical Technology, Institute of Pharmacy, University of Innsbruck, Innrain 80/82, A-6020 Innsbruck, Austria; patrick.knoll@uibk.ac.at (P.K.); andreas.bernkop@uibk.ac.at (A.B.-S.)

**Keywords:** hydroxypropyl-β-cyclodextrin, thiolation, ocular, mucoadhesive, aqueous humour, dexamethasone, drug delivery

## Abstract

The goal of this study was the design and evaluation of a thiolated cyclodextrin providing high drug solubilizing and mucoadhesive properties for ocular drug delivery. Hydroxypropyl-β-cyclodextrin (HP-β-CD) was thiolated via a microwave-assisted method, resulting in a degree of thiolation of 33%. Mucoadhesive properties of thiolated HP-β-CD (HP-β-CD-SH) were determined via rheological measurements and ex vivo studies on isolated porcine cornea. Due to thiolation of HP-β-CD, a 2-fold increase of mucus viscosity and a 1.4-fold increase in residence time on isolated corneal tissue were achieved. After instillation, the mean precorneal residence time and AUC of dexamethasone (DMS) eye drops were 4-fold and 11.7-fold enhanced by HP-β-CD-SH, respectively. Furthermore, in the presence of HP-β-CD-SH, a constant high level of DMS in aqueous humour between 30 and 150 min after administration was observed. These results suggest that HP-β-CD-SH is an excellent excipient for ocular formulations of poorly soluble drugs in order to prolong their ocular residence time and bioavailability.

## 1. Introduction

The therapeutic efficacy of many active pharmaceutical ingredients being in use to cure eye diseases is limited by a poor bioavailability that is often less than 5%. The main reason for this poor ocular bioavailability is a short drug residence time of a few minutes in the precorneal region, that is simply too short for absorption or for a therapeutic effect [1]. Therefore, most products on the market require frequent daily administrations to maintain therapeutic concentrations at the site of action, which entail not only discomfort in patients, but also risks of overdose and severe related side effects. Other drugs do not even reach the market at all as, due to their short precorneal residence time, the demanded therapeutic effect cannot be achieved. Strategies to prolong the precorneal residence time are primarily focusing on the use of mucoadhesive polymers. These auxiliary agents, however, cause an impaired vision, do not properly distribute over the ocular surface when used in higher concentrations and cannot improve the solubility of hydrophobic drugs. In order to address these shortcomings of mucoadhesive polymers, thiolated cyclodextrins were introduced as an ‘invisible choice’ to prolong ocular residence time of poorly soluble drugs [2]. In addition to the well-established advantages of cyclodextrins (CD) for ocular use, such as their ability to form water-soluble complexes with hydrophobic drugs [3], to enhance drug absorption into the eye, to improve aqueous stability and to reduce local irritation [4]; thiolated CD are able to form disulfide bonds with cysteine-rich subdomains of mucus glycoproteins providing comparatively high mucoadhesive properties. So far, however, evidence for these improved mucoadhesive properties on the ocular surface has just been provided via in vitro studies.

Therefore, the aim of this study was to thiolate a CD by utilizing a microwave (MW)-assisted method and to provide evidence for its potential as a mucoadhesive excipient for ophthalmic formulations via in vivo studies in rabbits. CD hydroxypropyl-β-cyclodextrin (HP-β-CD) was chosen, as it is the most commonly applied CD in ophthalmic products due to its high water solubility and high safety profile [5]. Dexamethasone (DMS), serving as a poorly soluble hydrophobic model drug, was incorporated in thiolated hydroxypropyl-β-cyclodextrin (HP-β-CD-SH). DMS was also chosen as a model drug because it is commonly used to treat inflammation of the anterior ocular segment. The ability of HP-β-CD-SH to increase the residence time of DMS in the tear fluid, and its bioavailability in the aqueous humour of rabbit eyes were evaluated.

## 2. Results and Discussion

### 2.1. Synthesis and Characterization of Thiolated Hydroxypropyl-β-cyclodextrin

Traditionally, chemical syntheses are conducted with conductive heating through external heat sources. This method takes a long time and is often inefficient because a high thermal conductivity of materials is required to transfer heat to the reagents. The heating process by microwave is very different from the traditional one. The microwaves interact directly with the molecules present in the reaction mixture, leading to an instantaneous and localized increase in temperature for anything that is subjected to dipolar rotation and ion conduction, i.e., the two fundamental mechanisms of heat conduction from microwave source to substances [6]. 

Several combinations of solvents and microwave conditions were tested to find optimized reaction conditions (Appendix A). Temperature, power and the solvent mixture are fundamental parameters in the MW-assisted synthesis. When temperature and power were increased, thiolation was more efficient. Furthermore, the more acidic the reaction mixture was, the higher was the degree of thiolation. An increase in temperature and/or microwave radiation and low pH, however, caused partial degradation of HP-ß-CD. Therefore, a compromise had to be found to optimize the reaction, such as an initial high temperature (i.e., 87 °C) for five minutes followed by a lower temperature (i.e., 80 °C) for the next 50 min. Regarding the effect of the acidic environment, mainly two kinds of solutions, either based on HCl or acetic acid, have been tested. With the first solution type, the thiolation degree was much higher; however, the second type was used as it allowed a higher yield. 

The MW–mediated synthesis in all cases comprises the first step of 45–55 min, during which the thiourea molecule attaches the glucopyranose unit of the CD, and a second hydrolysis step, during which the thiouronium ion turns into a thiol group bound to the CD as shown in Appendix A. The reaction is a SN2–type nucleophilic substitution, i.e., a concerted mechanism that involves the attachment of thiourea, a nucleophilic species, to a protonated primary alcohol group in acidic conditions. Thus, a molecule of H_2_O leaves, allowing the formation of a thiouronium ion, which is then hydrolyzed into a thiol with 10 M NaOH.

A method based on size exclusion chromatography was developed for the purification of HP-β-CD-SH. As seen in Appendix A, it was verified that the fractions 2, 3 and 4 contain HP-β-CD-SH, while the aliquots from 5 to 10 contain urea and thiourea. After purification, the yield in HP-β-CD-SH was 25%.

The product obtained was characterized by ^1^H NMR in DMSO-d_6_ and D_2_O, and the determination of the number of thiol groups was characterized by means of the Ellman’s modified assay. Appendix A shows the ¹H NMR spectrum in DMSO-d6 of HP-β-CD-SH, which can be compared with that of HP-β-CD (Appendix A). The signals at 1 ppm in Appendix A can be attributed to the protons of the hydroxypropyl group and the signal at 4.7 ppm to the anomeric C1. As it appears in Appendix A, following the reaction in the MW, the spectrum shows a reduction in the intensity of the peaks at 4.4 ppm and 5.7 ppm. This is a proof of the thiolation occurring on C6 and C2, respectively. In addition, the presence of a new peak is observed at about 1.96 ppm, which is attributable to the protons of the thiol group (-SH), confirming the thiolation in C6. As the hydroxyl group bound to C6 is a primary alcohol, it is more reactive with electrophilic reagents. In addition, the primary alcohol causes a steric hindrance, reducing the chances of substitution on the secondary alcohol [7].

To determine the degree of substitution with thiol groups, the peaks in the range of 5.6–4.9 ppm attributed to the anomeric protons were taken into consideration (see Appendix A). Among them, the two most shielded ones (H_A_: 5.03 ppm and H_B_: 5.169 ppm) are assigned to the anomeric of the native HP-β-CD, replaced or not with the hydroxypropyl group. The other three, the most deshielded peaks (H_C_: 5.34 ppm, H_D_: 5.45 ppm and H_E_: 5.61 ppm), are instead attributed to thiolated cyclodextrin with or without the hydroxypropyl functionality substituted in C6 or C2. By comparing the integrated areas of anomeric protons present in HP-β-CD-SH (unit H_C_ + H_D_ + H_E_ in Appendix A)) with those present in HP-β-CD (unit H_A_ + H_B_ in Appendix A), a thiolation degree of 33%, corresponding to 2.31 sulfur moieties per cyclodextrin molecule, was calculated. The Ellman’s test confirmed the calculated degree of substitution, showing a total thiol-group content of 150 ± 50 μmol/g corresponding to 0.5% wt. 

Compared to other thiolation methods found in the literature, refs. [8,9] the proposed synthesis is faster, free from halogenated activators and organic solvents, and, as it is MW assisted, more suitable for scale-up applications. The yield after purification was comparable to that obtained by Laquintana et al. [9], who used a traditional approach to prepare thiolated HP-β-CD. The lower thiolation degree obtained here should be weighed against the advantage of maintaining the solubility characteristics of the native HP-β-CD.

### 2.2. HP-β-CD-SH Mucoadhesivity Determination

The ability of mucoadhesive macromolecules to increase mucus viscosity via interactions, first described by Mortazavi and Smart [10], was since then reported in numerous studies [11,12]. The mucus is a gel layer, the viscoelastic properties of which are influenced by non-covalent interactions between mucin chains. The gel-strengthening associated with mucoadhesion depends on both the ability of the mucoadhesive polymer to penetrate into the mucus network and the strength of the interaction between the macromolecule and the mucus chains. 

HP-β-CD and HP-β-CD-SH were evaluated for their mucoadhesive potential. As previously stated, thiomers are able to form covalent bonds through oxidative coupling of thiol moieties and/or thiol/disulfide exchange reaction of the thiol groups present on oligomer chains and cysteine-rich subdomains of mucins [13]. The formation of these disulfide bonds can result in enhanced viscosity. Moreover, it has been shown that an increase in viscoelastic properties of mucus/oligomer mixtures directly correlates with mucoadhesive properties, suggesting an improved mucoadhesion.

#### 2.2.1. Viscosity Tests

To assess the mucoadhesivity of the CD under study by viscosity tests in vitro, porcine intestinal mucus was used instead of ocular mucus. Since the composition of the mucus present on the different mucous membranes is very similar, the use of porcine intestinal mucus has the advantage of allowing a large amount being collected for use in all in vitro studies [14]. 

As reported in the literature, thiolated CDs behave differently depending on their degree of thiolation. Highly thiolated CDs carry more than one thiol group per cyclodextrin and, therefore, are able to crosslink with the mucin network [7]. On the other hand, less thiolated CDs, i.e., those bearing one or less thiol group per CD unit, can be considered mucolytic compounds, just like N-acetylcysteine, as they break disulfide bonds via thiol/disulfide exchange reaction within the mucus [12].

Since the viscosity of the solutions of unmodified CD is not significantly different from that of the mucus/buffer mixture used as a control, it can be stated that unmodified CD is in fact non-mucolytic. 

As illustrated in Figure 1, HP-β-CD-SH increased mucus viscosity 2.2-fold within 60 min, whereas no viscosity improvement was registered within the first 30 min. Since HP-β-CD-SH carries about two thiol groups per CD molecule, it can likely crosslink with the mucin network.

#### 2.2.2. Microrheological Tests

The mucoadhesive properties of HP-β-CD-SH were also evaluated by micro-rheology analysis in comparison with plain HP-β-CD. As shown in Figure 2, there are no differences between the complex viscosity (η*) of the simple mucin dispersion and that of the mucin dispersion in the presence of HP-β-CD at time 0. On the contrary, η* of HP-β-CD-SH increased within one hour. The increase in complex viscosity is reflected in an increase of the modules G′ and G″. In particular, the increase in G′ confirmed the development of interconnected microstructures between mucin and HP-β-CD-SH [15]. These interconnections seem to occur with mucus glycoproteins, exhibiting the cysteine-rich subdomains typically involved in intra- and inter-molecular disulfide bridges [16].

#### 2.2.3. Ex-Vivo Tests 

The outcome of the corneal mucoadhesion study, shown in Figure 3, confirmed the results obtained in the rheological investigations. After 3 h, more than 85% of the HP-β-CD-SH still remained on the ocular surface, whereas just 60% of the unmodified CD remained there. This 1.4–fold improvement in residence time can be explained by the formation of disulfide bonds of thiolated CD with mucin glycoproteins [17].

### 2.3. Determination of the Dexamethasone/CD Association Constant

The ability of HP-β-CD and HP-β-CD-SH to form inclusion complexes with dexamethasone (DMS) was evaluated according to the Higuchi–Connors method [18]. As can be seen from Figure 4, the solubility of DMS increases as the concentration of the complexing agent (HP-β-CD or HP-β-CD-SH) increases. The intrinsic solubility of DMS under the experimental conditions is 0.2 mM. The association constant values for the DMS/HP-β-CD and DMS/HP-β-CD-SH complexes were found to be 932 ± 21 M^−1^ and 730 ± 17 M^−1^, respectively, while the complexation efficiency values were 0.34 and 0.22, respectively. Both HP-β-CD and HP-β-CD-SH were able to host DMS very efficiently and to increase its apparent solubility in aqueous media. Thiolation of HP-β-CD had obviously no significant effect on its ability to host DMS.

### 2.4. Cell Viability Assay 

In order to assess the toxicity of native HP-β-CD and HP-β-CD-SH, a cytotoxicity screening was performed using cell line BALB/3T3 clone A31, according to ISO 10993-5 (ISO 10993) for the biological evaluation of medical devices. The results of the cytotoxicity studies by the WST-1 assay are illustrated in Figure 5. Data obtained with HP-β-CD showed high cell viability in the concentration range of 1–20 mg/mL and estimated half-maximal inhibitory concentration (IC50) of 66 mg/mL. The HP-β-CD-SH sample showed to be non-cytotoxic in the range of 1–11.25 mg/mL. Its IC50 value was determined as 37 mg/mL. These data are in agreement with the already reported results obtained with Caco-2 cells [6,8,9]. These data demonstrate that HP-β-CD-SH although more toxic than HP-β-CD can be used safely.

### 2.5. In Vivo Studies

Male New Zealand albino rabbits are the most widely used animal model to evaluate ocular drug performance as their eyes have many anatomical and physiological similarities to human eyes [19]. 

The eye drops containing cyclodextrin/DMS complexes (DMS/HP-β-CD and DMS/HP-β-CD-SH) consisted of a 12.5% solution of HP-β-CD or HP-β-CD-SH, which was found to be non-toxic and non-irritating to rabbit eyes [20,21] and capable of solubilizing 0.3% DMS. A suspension of 0.3% DMS was used as control. 

#### 2.5.1. Draize Test

The modified Draize test showed a slight redness of the conjunctiva as illustrated in Figure 6, but neither secretion nor conjunctival edema during the first 30 min after applying the eye drops containing 0.3% of DMS. With the eye drops containing HP-β-CD or HP-β-CD-SH, no redness, secretions or conjunctival edema were detected after longer times from administration. The total I_irr_ score for each eye drop formulation turned out to be less than three. Hence it can be assumed that HP-β-CD and HP-β-CD-SH are endowed with adequate ocular tolerability.

#### 2.5.2. Kinetics of DMS Elimination from Tear Fluid 

DMS concentration in tear fluid (C_TF_) versus time profiles obtained with the different formulations tested were used to calculate the mean residence time (MRT) of DMS in tear fluid, according to the relevant non-compartmental theory [22]. MRT is obtained from the ratio between the area under the momentum curve, C_TF_*t vs t (AUMC), and the area under the curve, C_TF_ vs. t (AUC) [23].

In addition to MRT relating to each curve, Table 1 also shows the maximum residence time of the drug at quantifiable concentrations in tear fluid (RT_max_). This time corresponds to the last point of the C_TF_ vs time curve reported in Figure 7. Data in Table 1 show that HP-β-CD-SH increased the MRT by four times and the RT_max_ by six times compared to plain cyclodextrin. This can be explained by HP-β-CD-SH exhibiting high mucoadhesive properties because of its thiol groups forming disulfide bonds within the mucus with cysteine-rich subdomains of mucins via thiol/disulfide exchange reactions and/or via the oxidation of thiols [17]. These high mucoadhesive properties of HP-β-CD-SH and its ability to incorporate and dissolve DMS seem to strongly improve the therapeutic efficacy of this drug. 

#### 2.5.3. Measurement of DMS Intraocular Penetration 

Figure 8 shows the DMS concentration profiles in aqueous humour versus time, following the administration of a 0.15 mg dose via the control eye drops, DMS/HP-β-CD or DMS/HP-β -CD-SH. The corresponding values of area under curve (AUC), area under curve relative to that of the control (AUC_rel_), maximum concentration reached in the aqueous (C_max_) and time to reach this concentration (t_max_) are listed in Table 2. The DMS/HP-β-CD-SH formulation showed an AUC and C_max_ in aqueous humour 3- and 10-times higher than the control, respectively. Certainly, this can be attributed to the ability of HP-β-CD to maintain DMS in solution in the lacrimal fluid, at the absorption site, and also to its absorption enhancer features [24]. HP-β-CD was shown to extract cholesterol from corneal tissue, thus decreasing the resistance of the cell membrane to drug permeation [25]. Moreover, HP-β-CD did not significantly increase MRT and RT_max_ of DMS in tear fluid.

The DMS/HP-β-CD-SH formulation appears to increase the AUC in the aqueous humour 12 times compared to the control, and 4 times compared to the DMS/HP-β-CD formulation (Table 2). This observation can be explained by the ability of HP-β-CD-SH to promote DMS absorption. In fact, it has recently been demonstrated that thiolated β-CD is able to increase the P_app_ of sodium fluorescein through the cornea by about five times compared to an increase of only about two times by native β-CD [26]. However, the plateau of DMS concentration in the aqueous humour in the 30–150-min interval observed in the presence of HP-β-CD-SH (Figure 8) cannot be explained by its ability to promote DMS absorption. In fact, in the aqueous environment of the tear film drugs can be released from the drug-CD complexes through the absorption of cell membrane lipids, such as cholesterol and phospholipids, by CD, which involves the simultaneous release of the host drug from the complex. Therefore, drug release from the complex with CD seems to take place in close contact with the corneal epithelium that is favored by the high mucoadhesive properties of HP-β-CD-SH. These high mucoadhesive properties provided also an increase in MRT of the DMS and RT_max_ values as listed in Table 1. The plateau phase observed for DMS/HP-β-CD-SH as shown in Figure 8 can be explained by the assumption that for each molecule of DMS that is set free from the complex in tear fluid and permeates through the cornea, a new DMS molecule is set free from the complex, thus giving rise to pseudo-zero order permeation kinetics. 

## 3. Materials and Methods

### 3.1. Materials

Hydroxypropyl-β-cyclodextrin (HP-β-CD) MW 1380 Da (SR: 0.6), dexamethasone (DMS), thiourea, fluorescein diacetate (FDA), Ellman’s reagent (2,2′-dinitro-5,5′-dithiobenzoic acid), l-cysteine hydrochloride monohydrate, tris(hydroxymethyl)aminomethane, Sephadex G-15, Type II porcine gastric mucin, Dulbecco’s Modified Eagle Medium (DMEM), calf bovine serum, a mixture of antibiotics consisting of an aqueous solution of penicillin (10,000 U/mL) and streptomycin (10,000 µg/mL), 10 mM phosphate buffer pH 7.3 without Ca^2+^ and Mg^2+^ (PBS_A_), and a trypsin-EDTA buffer solution containing 0.25% trypsin were all purchased from Sigma-Aldrich (Darmstadt, Germany). Fibroblast BALB/3T3 clone A31 cells (CCL-163) were obtained from American Type Culture Collection.

### 3.2. Thiolation of HP-β-CD via Microwave Irradiation

A total of 400 mg of HP-β-CD were dissolved in 2 mL of 10% acetic acid, while 2.14 g of thiourea were dissolved at 40 °C under stirring in 8 mL of HCl 0.44 M. Once both compounds were solubilized, the dissolved thiourea was added dropwise to the CD solutions [7]. The resulting mixture was irradiated in a microwave device (Microonde Biotage Initiator) with a temperature-controlled setting and maximum power set at 90 Watt. The irradiation was carried out for 5 min at 87 °C and for 50 min at 80 °C. After this first step, hydrolysis was initiated by the addition of 10 M NaOH, resulting in a slightly basic pH of 8–9. The resulting mixture was irradiated by MW for 3 min at 80 °C.

### 3.3. Purification of HP-β-CD-SH

The product was brought to dryness in an evaporator (BÜCHI, Italy) at 30–40 °C. The solid product obtained was subjected to five washings, each with 20 mL of acetone, each followed by a 20-min centrifugation (5000 rpm; room temperature). The product was lyophilized (VirTis lyophilizer, freezing temperature −40 °C, drying at 30–40 mTorr, up to 16 °C). Purification was carried out by size exclusion chromatography, using a chromatographic column with a diameter of about 1.5 cm and a length of 30 cm, packed with Sephadex G-15 resin (22 cm). Milli-Q water was used as the mobile phase. A total of 1 mL of solution containing the synthesis product (100 mg/mL) was loaded onto the column. Twenty aliquots of 3 mL each were collected and analyzed by UV at 205 nm. The aliquots 2, 3 and 4 were combined and freeze-dried. The lyophilisate was subjected to a second purification in the size exclusion column, as already described. This time, only fraction 2 was retained.

### 3.4. NMR Characterization

NMR measurements were performed using a Brucker 400 UltraShieldTM spectrometer operating at 400 MHz for the 1H core. During the acquisition, the temperature (18 °C) was controlled by means of the Varian control unit (accuracy + 0.1 °C). The samples were dissolved in DMSO-d_6_ or D_2_O both at a concentration of 10 mg/mL. 

### 3.5. Determination of Thiol Content in HP-β-CD-SH

In order to calculate the amount of thiol groups present on HP-β-CD-SH, cysteine was used as standard. To reduce the existing intra- and inter-molecular disulfide bonds, the HP-β-CD-SH solutions were treated with NaBH_4_, then the amount of thiol groups was determined photometrically using Ellman’s reagent (5,5′-dithio-bis (2-nitrobenzoic acid)). The results obtained were expressed as μmol of thiol moieties per gram of product (μmol/g) [7]. 

### 3.6. Mucoadhesivity Determination of HP-β-CD-SH 

#### 3.6.1. Viscosity Tests

The dynamic viscosity of HP-β-CD and HP-β-CD-SH was measured in the presence of freshly excised porcine intestinal mucus. The porcine mucus was taken from the small intestine of a freshly slaughtered pig, donated by a local slaughterhouse and purified as previously described [27]. Dynamic oscillatory tests were performed at a frequency of 1 Hz on a plate-plate viscometer (Haake MARS Rheometer, 379-0200; Thermo Electron GmbH, Karlsruhe, Germany). In detail, CD solutions (0.1% *w/v*) in 200 mM phosphate buffer were prepared. A total of 250 µL of the solution at pH 7 were mixed with 750 mg of porcine intestinal mucus. The mixture was incubated for 0.5 or 1 h at 37 °C, and then transferred to the rheometer plate for measurement. All the measurements were carried out in triplicate.

#### 3.6.2. Microrheological Tests

The measurements were performed using the Nano Zetasizer ZS instrument as previously reported [15], and followed the theory reported by Dodero [28]. The microrheological characterization of HP-β-CD and HP-β-CD-SH was performed using type II porcine gastric mucin. As ocular mucins are not commercially available, porcine gastric mucin was used, as it has already been used as a model in other studies concerning ocular mucoadhesion [29]. Briefly, a 3 mg/mL suspension of type II porcine gastric mucin was prepared in Milli-Q water and filtered (0.45 µm cellulose acetate filters). A solution of HP-β-CD or HP-β-CD-SH was then prepared in deionized water in a concentration of 10 mg/mL. An amount of 1 mL of each sample tested had the following composition: 5 µL/mL of tracer (latex polystyrene particles, diameter 500 nm, Beckman), 7.38 mg/mL of HP-β-CD or HP-β-CD-SH, 0.3 mg/mL of mucin and 8.1 mg/mL of 5% NaCl. As a reference, 1 mL of sample was prepared having the same composition but without HP-β-CD or HP-β-CD-SH. All measurements were carried out in triplicate.

#### 3.6.3. Ex Vivo Tests

For these tests porcine eyes were freshly obtained from a local slaughterhouse. HP-β-CD and HP-β-CD-SH were fluorescent-labelled by incorporating FDA for assessment of mucoadhesive properties, as previously described [30]. Briefly, 100 mg of CD were dissolved in 100 mL of deionized water and the pH of the solution was adjusted to 6.5 with 100 mM HCl. Then, 5 mg of FDA were dissolved in 10 mL of 96% ethanol and 5 mL of this solution were added to each CD solution. After 24 h of stirring at room temperature, the suspensions were filtered (25 mL syringe, 22 μm syringe cellulose filter) in order to eliminate free FDA, and then freeze dried. 

To evaluate ex vivo the mucoadhesive properties of HP-β-CD or HP-β-CD-SH, a previously reported method was used [30]. In brief, solutions of 20 mg of FDA-labelled HP-β-CD or HP-β-CD-SH in 100 μL of simulated tear fluid (STF, containing 1.7893 g/L potassium chloride, 6.3118 g/L sodium chloride, 2.1842 g/L sodium bicarbonate, 44.4 mg/L calcium chloride and 47.6 mg/L magnesium chloride, pH 7.4) [31] were prepared. Each sample was added dropwise on the corneal mucosal surface of the excised eye, previously lodged in a PP tube. The tubes were held horizontally for 10 min, after which STF flow (1 mL/min) was restarted. The STF flowing down the mucosa was collected after 30, 60, 90, 120, 150 and 180 min and analyzed for the FDA hydrolyzed to sodium fluorescein [12] by the microplate reader (M200 spectrometer; Tecan infinite, Grödig, Austria) at an emission wavelength of 535 nm and exciting wavelength of 485 nm. All experiments were performed in triplicate.

### 3.7. Determination of the DMS/Cyclodextrins Association Constant

For this analysis, stock solutions of HP-β-CD or HPβ-CD-SH in 37.5 mM phosphate buffer pH 7.4 (PB) (20 mg/mL) were prepared. Samples of 1 mL each were prepared containing increasing concentrations (1–20 mg/mL range) of HP-β-CD or HP-β-CD-SH in PB: a total of 1 mg of DMS was added to each sample and placed in a shaking water bath at 25 °C for two hours. They were then centrifuged twice, each time for 15 min at 10,000 rpm. The supernatant of each sample was analyzed by UV at the wavelength of 242 nm. The association constant was calculated using the method of Higuchi and Connors [18].

### 3.8. Cell Viability Assay

The fibroblast BALB/3T3 clone A31 cell line was employed in this study. Fibroblast BALB/3T3 clone A31 cells were grown in Dulbecco’s Modified Eagle’s Medium (DMEM), supplemented with 2 mM L-glutamine, 1% penicillin/streptomycin and 10% calf bovine serum. Cells were grown in a CO_2_ incubator (Heracell 150i series) at 37 °C and 5% CO_2_, with cells subcultured at 80–90% confluency. Cell monolayers were rinsed with PBS and treated with trypsin-EDTA to detach cells before resuspension in fresh media. A subconfluent monolayer of fibroblast BALB/3T3 clone A31 cells was trypsinized, centrifuged at 1000 rpm for 5 min, resuspended in growth medium, and then counted. Fibroblast BALB/3T3 clone A31 cells were seeded in each well of 96-well plates at a seeding density of 8 × 10^4^ cells/well for 24 h until 60–70% confluence was reached. After 24 h, the medium was removed from each well and replaced with DMEM containing HP-β-CD or HP-β-CD-SH in a concentration range from 0.1% to 5% (*w/v*) to evaluate their cytotoxicity. After 4 h of incubation, media were removed and substituted with fresh medium containing 10% WST-1 reagent solution and maintained for 4 h at 37 °C, 5% CO_2_. Afterwards, formazan dye absorbance was quantified at 450 nm with the reference wavelength at 655 nm by using a multilabel reader (BioTek 800/TS, Thermo Scientific, Cheshire, UK).

### 3.9. In Vivo Studies

Male New Zealand albino rabbits weighing approximately 2.6–3 kg, kept in standard housing conditions, were used for in vivo studies. The animals were treated as prescribed in the guidelines from the European Community Council Directive 2010/63. The animal protocol was approved by the Italian Ministry of University and Research (authorization n. 192/2019-PR). The samples tested had the following compositions:0.3% (*w/v*) DMS (control);0.3% (*w/v*) DMS containing 12.5% (*w/v*) of HP-β-CD (DMS/HP-β-CD);0.3% (*w/v*) DMS containing 12.5% (*w/v*) of HP-β-CD-SH (DMS/HP-β-CD-SH).

#### 3.9.1. Draize Test

To assess the irritation of rabbit eyes after instillation of the eye drops under study, a modified Draize test was performed following the procedure described by [32]. This irritation test was conducted using a 0 (absence) to 3 (maximum) clinical rating scale for lachrymation, edema and conjunctival redness [33]. For each of the samples under study the test was performed on three rabbits, through a single instillation of the eye drops (50 μL) per rabbit eye, while the untreated contralateral eye was used as control. Each animal was observed at 0.5, 1, 2, 6 and 24 h after instillation. An overall irritation index (I_irr_) was calculated by adding the total clinical evaluation scores for each observation time. A score of 2 or 3 in any category or an I_irr_ greater than 4 is considered an indicator of a clinically significant irritation.

#### 3.9.2. Determination of DMS Elimination Kinetics from the Tear Fluid

To determine the DMS elimination kinetics from tear fluid, one drop of each sample under study (50 µL), corresponding to 0.15 mg of DMS, was instilled by a Gilson pipette into the lower conjunctival sac of each rabbit, avoiding spillage.

At predefined intervals, tear fluid samples were taken from the lower margin strip using 1 µL disposable glass capillaries (Microcaps, Drummond Scientific Co., St. Louis, MO, USA) which were washed with 1 µL of phosphate buffer (PBS; pH 7.4, 0.0375 mM). Each sample was diluted with 100 µL of PBS and was injected directly for analysis by HPLC.

For each of the three formulations studied, elimination curves were constructed, each determined from a single eye of different rabbits, by taking tear fluid samples at 2, 4, 6, 8, 10, 15, 20, 25, 30, 40, 50, 60, 75, 90 and 120 min from instillation.

#### 3.9.3. DMS Pharmacokinetics in the Aqueous Humour

For measuring DMS pharmacokinetics in the aqueous humour, one drop of each eye drop formulation under study was instilled in the rabbit eye. During the experiments, the rabbits were placed in restraining boxes, where they could move their heads and eyes freely. After a pre-established time from instillation, namely, 30, 45, 60, 90, 120, 150 or 180 min each rabbit eye was anaesthetized by instilling one drop of Novesina^®^, then 60–80 μL of aqueous humour was aspirated from the anterior chamber, using a 1.0-mL insulin syringe fitted with a 29-gauge needle. The single eyes of at least six animals were used for each data point. In order to limit the number of animals, each eye was randomly re-used, after a 2-week interval at least three times. For analysis, each sample was mixed with an equal volume of acetonitrile, which was then centrifuged, and 20 μL of the supernatant were analyzed by HPLC as previously described [34,35]. 

### 3.10. Statistical Data Analyses

For the DMS pharmacokinetics in the aqueous humour, the linear trapezoidal rule between 0 and 180 min was used to calculate the area under curve (AUC) and the statistical differences were evaluated using the method reported by Schoenwald et al. [36]. Student *t*-test was applied as *p* < 0.05 or *p* < 0.01 depending on the analysed property.

## 4. Conclusions

In this study, HP-β-CD-SH was synthesized using a microwave-assisted method. The thiolated CD showed excellent mucoadhesive properties and negligible cytotoxicity. Furthermore, the model drug DMS could be efficiently incorporated in this thiolated CD providing an improved aqueous solubility. In vivo studies with eye drops containing HP-β-CD-SH hosting DMS showed a prolonged residence time of DMS in the precorneal region of rabbits as well as a significant absorption enhancement, resulting in an almost constant concentration of DMS in the aqueous humour for two hours. According to these results, thiolated CD seem to be highly potent excipients in order to improve ocular bioavailability of drugs that sufferer from poor solubility in lacrimal fluid and a too short ocular residence time.

## Figures and Tables

**Figure 1 ijms-23-02612-f001:**
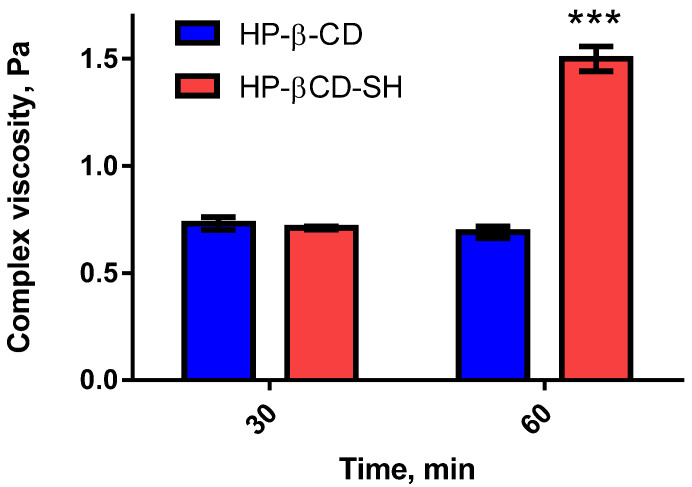
Viscosity of HP-β-CD or HP-β-CD-SH (0.1% *w/v*) with mucus measured on a plate-plate viscometer (1 Hz frequency). Each point is the mean ± standard deviation (SD) of at least three values (*** *p* < 0.001).

**Figure 2 ijms-23-02612-f002:**
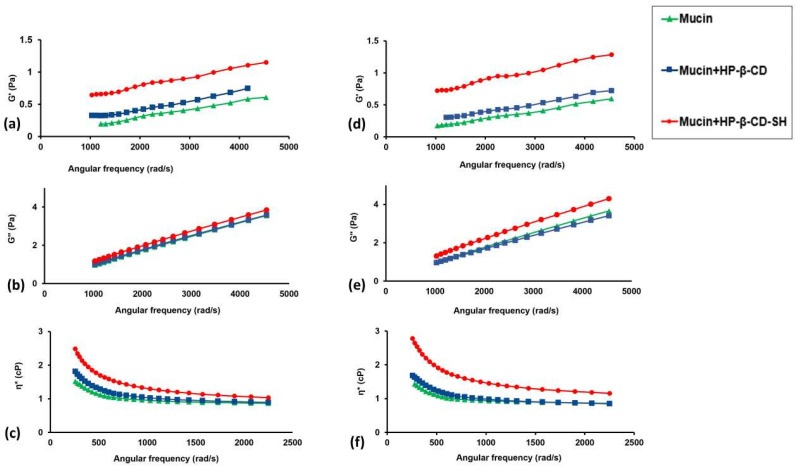
(**a**) Elastic modulus (G′), (**b**) viscous modulus (G″), (**c**) complex viscosity (η*) of HP-β-CD and HP-β-CD-SH with respect to the mucin dispersion at time 0. (**d**) Elastic modulus (G′), (**e**) viscous modulus (G″), (**f**) complex viscosity (η*) of HP-β-CD and HP-β-CD-SH with respect to the dispersion of mucin after 1 h.

**Figure 3 ijms-23-02612-f003:**
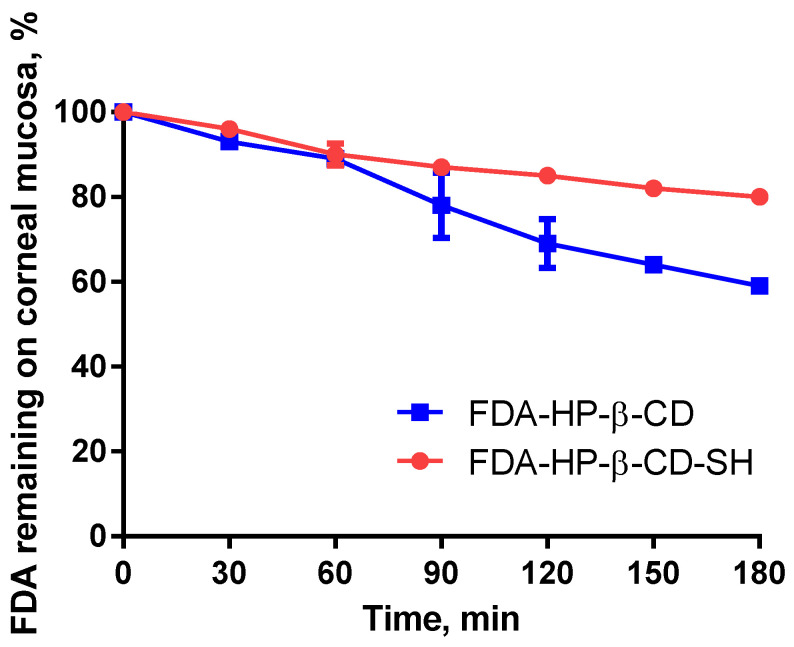
Time course of percentage of FDA incorporated in HP-β-SH or HP-β-CD remaining on corneal mucosa continuously rinsed with 100 mM phosphate buffer at 37 °C and 100% relative humidity. Each point is the mean ± SD of three values.

**Figure 4 ijms-23-02612-f004:**
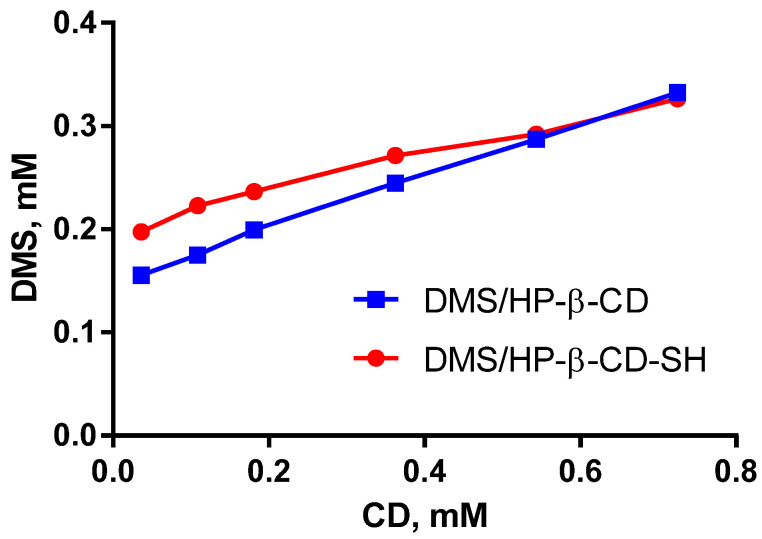
DMS solubility profiles in the presence of HP-β-CD or HP-β-CD-SH. Each point is the mean ± SD of 3 values.

**Figure 5 ijms-23-02612-f005:**
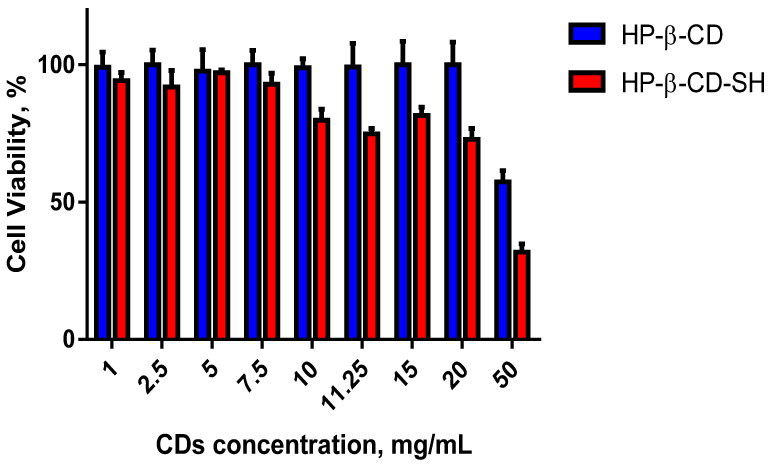
Cell viability screening performed on BALB/3T3 cell line clone A31, exposed for 4 h to HP-β-CD or HP-β-CD-SH in the 1–50 mg/mL concentration range. Untreated cells were used as control. The values indicated in the figure are means ± SD of 6 replicates.

**Figure 6 ijms-23-02612-f006:**
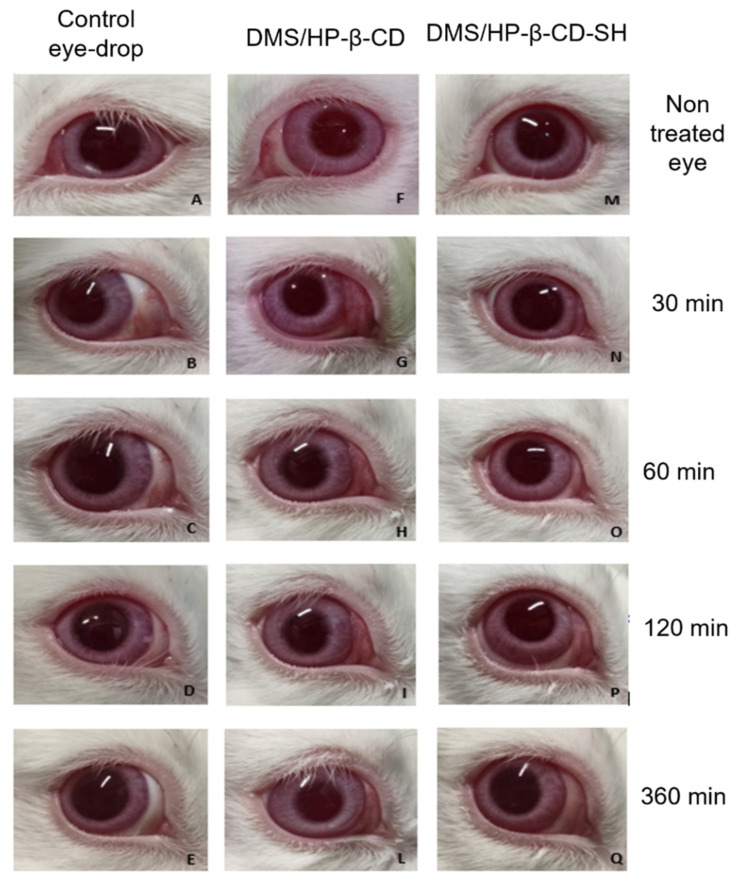
Modified Draize test representation. (**A**) Left untreated eye of rabbit used as reference; (**B**–**E**) the same rabbit right eye after instillation of one drop of eye drops containing 0.3% (m/V) DMS at times: 30 min, 1 h, 2 h and 6 h, respectively. (**F**) Left untreated eye of rabbit used as reference; (**G**–**L**) the same rabbit right eye after instillation of one drop of eye drops containing 0.3% (m/V) DMS and 12.5% HP-β-CD at the time: 30 min, 1 h, 2 h and 6 h, respectively. (**M**) Left untreated eye of rabbit used as reference; (**N**–**Q**) the same right rabbit eye after instillation of one drop of eye drops containing 0.3% (m/V) DMS and 12.5% HP-β-CD-SH at times: 30 min, 1 h, 2 h and 6 h, respectively.

**Figure 7 ijms-23-02612-f007:**
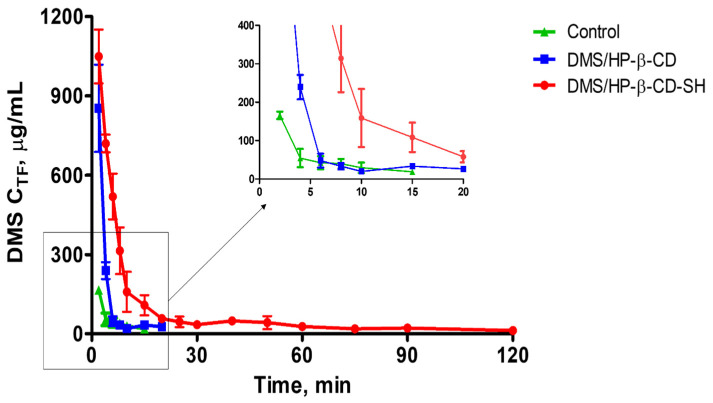
Profiles of DMS elimination from tear fluid of rabbits following instillation of one eye drop (50 μL) containing 0.3% *w/v* DMS alone or DMS and 12.5% *w/v* of different CDs. Each point is the mean ± SD of at least 6 values.

**Figure 8 ijms-23-02612-f008:**
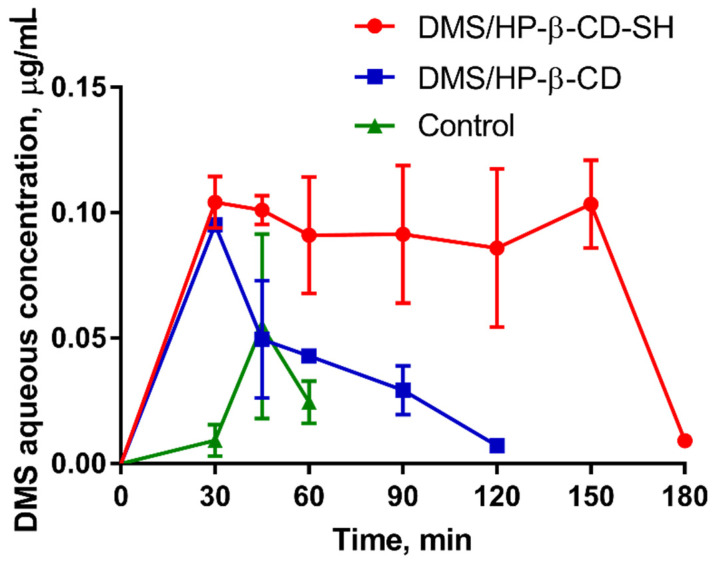
Profiles of DMS pharmacokinetics in the aqueous humour of rabbits following instillation of one eye drop (50 μL) containing 0.3% *w/v* DMS alone or DMS and 12.5% *w/v* of different CDs. Each point is the mean ± SD of at least six values.

**Table 1 ijms-23-02612-t001:** Effects of CDs on DMS residence time in tear fluid of rabbits after instillation of one ophthalmic drop (50 μL) containing 0.3% *w/v* DMS and 12.5% *w/v* of different CDs. MRT, mean residence time; RT_max_, maximum residence time at measurable concentrations (≥2.5 μg/mL) (*n* = 6).

Formulation	MRT ± SD (min)	RTmax (min)
Control	6.2 ± 0.6	15
DMS/HP-β-CD	5.1 ± 0.5 **	20
DMS/HP-β-CD-SH	24.4 ± 1.4 *^,^**	120

* Significantly different from control. ** Significantly different from each other (*p* < 0.05).

**Table 2 ijms-23-02612-t002:** Pharmacokinetic data in the aqueous humour following instillation of one eye drop (50 μL) containing 0.3% *w/v* DMS or 0.3% *w/v* DMS and 12.5% *w/v* of different CD.

Formulation	C_max_±SD, μg mL^−1^	T_max_, min	AUC±SD, μg mL^−1^ min	AUC_rel_
Control	0.055 ± 0.036	45	1.1 ± 0.4	-
DMS/HP-β-CD	0.095 ± 0.014 *	30	3.4 ± 0.3 *^,^**	3.1
DMS/HP-β-CD-SH	-	-	12.9 ± 1.0 *^,^**	11.7

Means ± SD of at least six values obtained with different animals. * Significantly different from control. ** Significantly different from each other (*p* < 0.05).

## Data Availability

Not applicable.

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
