# Peer review of "Thiolated Hydroxypropyl-β-cyclodextrin: A Potential Multifunctional Excipient for Ocular Drug Delivery"

_ijms, 2022, doi:10.3390/ijms23052612_

Round 1

Reviewer 1 Report

I only have 2 questions related to rabbit studies design, please address my questions below. 

  1. In method 3.9.2, please indicate the status of rabbits when the authors conducted eyedrops installation. Did you anesthesia the rabbits when you do the eyedrops administration? If the rabbits were knocked down, the eye blinking rate can be reduced which can influence the clearance of the eyedrops from the ocular surface.
  2. In method 3.9.3, the author indicated that the rabbit eyes were reused. Please provide more detailed information about how these rabbit eyes were reused, which groups shared the same rabbits. Actually, for rabbit eyes, one time collecting 60~80 µl aqueous humor is not less and it may need more than 2 weeks for the aqueous humor recovery. If the aqueous humor volume cannot recovery to their original status, then the drug concentration at later time points may not be accurate.

Author Response

Reviewer #1   

In method 3.9.2, please indicate the status of rabbits when the authors conducted eyedrops installation. Did you anesthesia the rabbits when you do the eyedrops administration? If the rabbits were knocked down, the eye blinking rate can be reduced which can influence the clearance of the eyedrops from the ocular surface.

Reply: “During the experiments the rabbits were placed in restraining boxes, where they could move their heads and eyes freely.” This sentence was added to the method section.

    In method 3.9.3, the author indicated that the rabbit eyes were reused. Please provide more detailed information about how these rabbit eyes were reused, which groups shared the same rabbits. Actually, for rabbit eyes, one time collecting 60~80 µl aqueous humor is not less and it may need more than 2 weeks for the aqueous humor recovery. If the aqueous humor volume cannot recovery to their original status, then the drug concentration at later time points may not be accurate.

Reply: The sentence has been changed as follows: “In order to limit the number of animals, each eye was randomly re-used, after a 2-week interval at least 3 times”.

Reviewer 2 Report

The manuscript submitted for review in the journal IJMS describes the synthesis of a thiolated hydroxypropyl-β-cyclodextrin derivative and a number of its physicochemical and biological properties. The synthesis was thoroughly optimized and the product obtained was properly analyzed. Such a wide choice of biological tests (in vivo and ex vivo) is noteworthy. 

Since the obtained derivative is not yet used as an excipient in ocular drugs, it would be useful to add the word "potential" in the title of the article (or other suitable). 

It would be useful to give more justification for the choice of dexamethasone as a model drug. Simply stating that it is poorly soluble and hydrophobic is not enough. Most ocular drugs have just such properties. 

The presentation of the results of association constants (l. 190) should be improved. Everywhere else the results with uncertainties are given correctly. Here there is an error. The uncertainty can have up to two significant digits. The result and the uncertainty should have the same decimal expansion. Uncertainties are always rounded up. 

Detailed remarks
Materials

What does "tris (hydroxymethyl) pure aminomethane" mean? (l. 307 and 308).

It is: 5.000 rpm. Should be: 5,000 rpm. (l. 326).
 Better to write: 10,000 rpm. (l. 401).
Supplementary material
Table S1 and S2
It is: Thourea. Should be: Thiourea.
It is: T. Should be: t. Symbol 'T' is reserved for thermodynamic temperature (in K) not for temperature in Celsius degrees.
Too much precision in values of yield. One digit after period is far enough.
It is: Vassel. Should be: Vessel.
There is no need to follow the values of yield by "%" because this unit is depicted at the beginning of a column.
Figure S1
The scheme should have higher resolution. Moreover, the 'OR' notations are missed at C3 and C6. It should be as in the latter structures.

Author Response

The manuscript submitted for review in the journal IJMS describes the synthesis of a thiolated hydroxypropyl-β-cyclodextrin derivative and a number of its physicochemical and biological properties. The synthesis was thoroughly optimized and the product obtained was properly analyzed. Such a wide choice of biological tests (in vivo and ex vivo) is noteworthy.

Since the obtained derivative is not yet used as an excipient in ocular drugs, it would be useful to add the word "potential" in the title of the article (or other suitable).

Reply: The title has been changed as follows: “Thiolated hydroxypropyl-β-cyclodextrin: A potential multifunctional excipient for ocular drug delivery”

It would be useful to give more justification for the choice of dexamethasone as a model drug. Simply stating that it is poorly soluble and hydrophobic is not enough. Most ocular drugs have just such properties.

Reply: “DMS was chosen as model drug because it is commonly used to treat inflammation of the anterior ocular segment.”. This sentence was added to the introduction section.

The presentation of the results of association constants (l. 190) should be improved. Everywhere else the results with uncertainties are given correctly. Here there is an error. The uncertainty can have up to two significant digits. The result and the uncertainty should have the same decimal expansion. Uncertainties are always rounded up.

Reply: The decimal expansion of the standard error has been rounded up.

Detailed remarks

Materials

What does "tris (hydroxymethyl) pure aminomethane" mean? (l. 307 and 308).

Reply: The word tris (hydroxymethyl) pure aminomethane has been replaced with tris(hydroxymethyl)aminomethane

It is: 5.000 rpm. Should be: 5,000 rpm. (l. 326). Better to write: 10,000 rpm. (l. 401).

Supplementary material

Table S1 and S2

It is: Thourea. Should be: Thiourea.

It is: T. Should be: t. Symbol 'T' is reserved for thermodynamic temperature (in K) not for temperature in Celsius degrees.

Too much precision in values of yield. One digit after period is far enough.

It is: Vassel. Should be: Vessel.

There is no need to follow the values of yield by "%" because this unit is depicted at the beginning of a column.

Figure S1

The scheme should have higher resolution. Moreover, the 'OR' notations are missed at C3 and C6. It should be as in the latter structures.

Reply: All these corrections have been made.